# ADVERSARIAL AND NATURAL PERTURBATIONS FOR GENERAL ROBUSTNESS

## ABSTRACT

In this paper we aim to explore the general robustness of neural network classifiers by utilizing adversarial as well as natural perturbations. Different from previous works which mainly focus on studying the robustness of neural networks against adversarial perturbations, we also evaluate their robustness on natural perturbations before and after robustification. After standardizing the comparison between adversarial and natural perturbations, we demonstrate that although adversarial training improves the performance of the networks against adversarial perturbations, it leads to drop in the performance for naturally perturbed samples besides clean samples. In contrast, natural perturbations like elastic deformations, occlusions and wave does not only improve the performance against natural perturbations, but also lead to improvement in the performance for the adversarial perturbations. Additionally they do not drop the accuracy on the clean images.

## 1 INTRODUCTION

A large body of work in computer vision and machine learning research focuses on studying the robustness of neural networks against adversarial perturbations (Kurakin et al., 2016; Goodfellow et al., 2014; Carlini & Wagner, 2017). Various defense based methods have also been proposed against these adversarial perturbations (Goodfellow et al., 2014; Madry et al., 2017; Zhang et al., 2019b; Song et al., 2019). Concurrently, research also shows that deep neural networks are not even robust to small random perturbations e.g. Gaussian noise, small rotations and translations (Dodge & Karam, 2017; Fawzi & Frossard, 2015; Kanbak et al., 2018). There is plenty of research being performed in the domain of adversarial perturbations however, there is very little focus on robustifying the networks against natural perturbations as we do here.

Furthermore, adversarial perturbations are difficult to be found in the real world, and naturally occuring perturbations are of different nature than these pixel based perturbations. Therefore, in this paper we consider natural perturbations of six different styles that are elastic, occlusion, Gaussian noise, wave, saturation, and Gaussian blur. In this, "elastic" denotes a random sheer transformation applied to the image, "occlusion " is a large randomly located dot in the image and "wave" is a random geometric distortion applied to the image. Additionally, there is no consensus about whether adversarial robustness helps against natural perturbations. Zhang & Zhu (2019) showed that adversarial training reduces texture bias. However, Engstrom et al. (2019) demonstrated that $l_\infty$ based robustness does not generalize to natural transformations like rotations and translations. Here we evaluate whether adversarial training helps against natural perturbations and vice versa.

Besides the robustness of the neural networks against natural and adversarial perturbations there is an open debate in the literature about the trade-off between the robustness and the accuracy (Tsipras et al., 2018; Zhang et al., 2019a; Su et al., 2018). Contrasting with adversarial training we found that networks partially trained with naturally perturbed images does not degrade the classification performance on the clean images. On the CIFAR-10 dataset, we even found that partial training with naturally perturbed images improves the classification accuracy for clean images.

Given that deep neural networks are on par in performance with humans or they even surpass humans on clean images however, they fail to perform well on small natural perturbations (He et al., 2016; Dodge & Karam, 2017). Hendrycks & Dietterich (2019) introduced a subset of Imagenet Deng et al. (2009) called Imagenet-C with corruptions applied on images from Imagenet. Although

in Imagenet-C each corruption has five severity levels however they are not standardized for comparison among them. We standardize the effect of perturbations on training data to a fixed drop in classification accuracy of the test set, through this we allow for a fair comparison between different styles of training to retain robustness in the classifier against perturbations. We also normalize the performance of the network on different datasets to compare the robustness of the network for different datasets.

We conduct 320 experiments on five different datasets for adversarial and six different natural perturbations. General robustness is the most desired case given as, the robustness against perturbations not seen during the training of the classifier. Hence, we evaluate the general robustness of networks by testing them on seen perturbations i.e. when the training and testing type of perturbations is the same, as well as on unseen perturbations i.e. when the training and testing type of perturbations are different. Among classifiers tested on the both seen and unseen perturbations, the natural perturbations of elastic, wave and occlusion come out on top compared to other natural perturbations as well as compared to adversarial perturbations. Our contributions are given as follows: i) We perform fair evaluation of robustness. ii) We show that, natural perturbation robust classifiers generalize to clean images. iii) We depict that, seen natural perturbations are more robust than seen adversarial perturbations. iv) Our evaluation for general robustness shows natural elastic, wave and occlusion perturbations are best robust against unseen perturbations.

## 2 RELATED WORK

**Robustness with Adversarial or Natural Perturbations.** In Goodfellow et al. (2014) the robustness of neural networks was demonstrated by adding imperceptible i.e. adversarial perturbations in the input to the degree that it will misclassify the input into the wrong class. To solve the problem Carlini & Wagner (2017) proposed adversarial training (AT) procedure that is by training the network on adversarially perturbed images networks can be robustified against these perturbations. In this work we employ a strong yet undefended attack "basic iterative method (BIM)" to generate adversarial examples. "Projected gradient descent (PGD)" a state of the art defense technique for adversarial training to evaluate its effectiveness compared to other ways of robustification. Zhang et al. (2019a); Tsipras et al. (2018) questioned the generalization capability of adversarially trained neural networks on the clean images and showed that with the increase in adversarial robustness the accuracy of the networks on clean images drops. Therefore, we evaluate the performance of adversarially trained networks on clean, adversarial as well as natural perturbations, and compare them with networks trained with natural perturbations.

Hendrycks & Dieterich (2019) focused on testing the robustness of vanilla neural networks on 15 different natural perturbations with different perturbation levels. We observe that some of their perturbations are correlated e.g. Gaussian noise, shot noise and impulse noise (Laugros et al., 2019). While, in our work we train and test on six different natural perturbations covering the breadth of styles of natural perturbations. Furthermore, instead of selecting different perturbation levels randomly we standardize their effect by dropping the accuracy of the network to a fix level for fair comparison among them. Finally, rather than testing vanilla networks, we propose to robustify the networks with natural perturbations and test them for both adversarial and natural perturbations.

**General Robustness with Adversarial and Natural Perturbations.** Ford et al. (2019) established the close connections between adversarial robustness and natural perturbations robustness and suggested that adversarial and natural perturbations robustness should go hand in hand and networks should be robustified against both of them. In another similar line of work Kang et al. (2019); Engstrom et al. (2019), proposed natural perturbations based adversarial attacks and showed that testing with only one type of adversarial perturbations does not tell about the complete robustness of the network. We focus on determining the general robustness of neural network classifiers by testing them against unseen adversarial and natural perturbations.

Rusak et al. (2020) focus on robustification against natural corruptions besides adversarial perturbations. They utilize Gaussian and speckle noise and show that by augmenting the properly tuned training of a network with Gaussian noise makes it generalizable to unseen natural perturbations. However, in this work we show that elastic, wave and occlusion perturbations surpass the robustness with Gaussian noise. Laugros et al. (2019) study the relationship between adversarial and natural

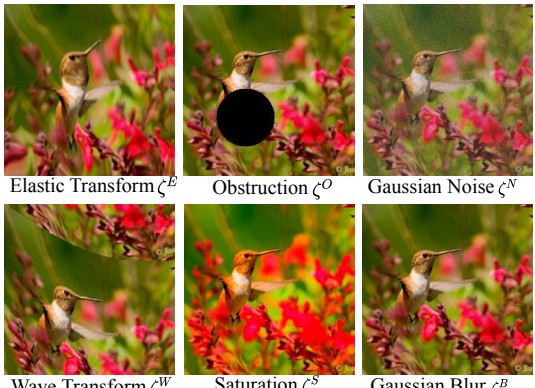

| Elastic Transform $\zeta^E$ | Obstruction $\zeta^O$ | Gaussian Noise $\zeta^N$ |
| Wave Transform $\zeta^W$ | Saturation $\zeta^S$ | Gaussian Blur $\zeta^B$ |

Figure 1: Randomly selected sample images of six different natural perturbations used in our experiments. Note that the perturbations for each image vary e.g. for another image the the occlusion will be at another position in the image.

perturbations. However, they do not study elastic and wave transforms. Furthermore, they generate adversarial examples by randomly selecting parameters but we select the parameters of both natural and adversarial perturbations by standardizing the effect of perturbations. So, their results contrast with ours. They also do not evaluate the performance of robustified networks on clean images.

## 3 METHOD

Given the $n_{th}$ input image $x_n \in \mathbb{R}^2$, and the output class $y_n \in \mathbb{N}$, a standard classifier $f(x_n) = y_n$ predicts the class. In the real world, inputs of the classifiers may deviate from the learning set, whose members will be referred to as clean images. As representatives of the perturbed images we consider sets of naturally occurring perturbations $\zeta_n^t$ and adversarial perturbations $\zeta_n^A$ for the purpose of enhancing the robustness of the classifier.

**Constructing Adversarial Perturbations.** Adversarial examples satisfy two properties 1) the class for the perturbed image is different from the class predicted for clean image i.e. $f(\zeta_n^A(x_n)) \neq f(x_n)$, 2) they are visually similar and their similarity is determined by the $l_p$- norm. While fulfilling these two properties we follow the standard procedure of the basic iterative method Kurakin et al. (2016) to introduce adversarial perturbations $\zeta_n^A(x_n)$ in the images by finding the perturbation $\delta_n$ with a small norm $l_\infty$ bounded by $\epsilon$ such that, $f(x_n) \neq f(\zeta_n^A(x_n))$, where $\zeta_n^A = x_n + \delta_n$ and $\delta_n \leq \epsilon$. The equation to be optimized is given as

$$\zeta_n^A(x_n^0) = x_n + \delta \tag{1}$$

$$\zeta_n^A(x_n^{k+1}) = \zeta_n^A(x_n^k) + \epsilon_s \text{Sign}(\bigtriangledown_x(\mathcal{L}_r^\delta(\zeta_n^A(x_n^k), y_n, \theta))) \tag{2}$$

where, $\epsilon_s$ is the step size at step $k$.

**Constructing Natural Perturbations.** For natural perturbations we also restrict them to satisfy two properties 1) the overall drop in the performance of a classifier is the same as the drop with the adversarial perturbations for comparison among them, 2) they are visually similar enough to be correctly classified by humans. We consider a set of naturally occurring perturbations $\zeta_n^t$, where $t \in \{E, O, N, W, S, B\}$ denotes the type of perturbation operator. We construct images $\zeta_n^t(x_n)$ by selecting a perturbation operator from $t$. When tested on a standard classifier, the perturbation will cause a drop in the performance. Selected samples of the six natural perturbations are shown in the Figure. 1.

The first natural perturbation is Elastic deformation $\zeta^E$. Elastic deformation appears in small changes in the viewing angles. We introduce this perturbation by $\zeta^E = \mathcal{T}(x_n, \alpha x_n' \circledast \mathcal{N}(\mu, \sigma^2))$, where, $x' \in \text{rand}(-1, +1)$, selects random number between $-1$ and $+1$, generated with a uniform distribution and $\mathcal{T}$ is the affine transform. Occlusion is created with $\zeta^O = \min(x_n, b^{x_c, t, r})$, where, $b$ is a matrix of zeros with $x_c$ as its center and $t$, $r$ being the thickness and radius of the circle respectively. Gaussian noise is introduced using $\zeta^N(x_n) = x_n + x_n^{N(\mu, \sigma^2)}$. A Wave transform is

introduced in the images through $\zeta^W = x_n \longmapsto (x_n + \sin(2\pi x_n w))$, where, $\longmapsto$ is a shift operator. Saturation is added by using $\zeta^S = (1 - \alpha)x' + \alpha x_n$, where, $\alpha \in [0, 1]$, $x'$ is the black and white version of $x_n$. Gaussian blur $\zeta^B$ is added by convolving a two dimensional Gaussian function to the image.

Although these natural perturbations are class agnostic however, they are image specific that is, the perturbation for each image is different. For elastic transform the intensity of the transform is varied, in the occlusion the position of occlusion is randomly selected for each image, intensity of Gaussian noise is varied uniformly at random, wave is also scaled uniformly at random for each image, similarly saturation factor is also uniformly selected and finally, the intensity of Gaussian blur is uniformly randomly sampled for each image.

**Fair Comparison.** To permit the fair comparison between natural and adversarial perturbations, instead of selecting perturbations randomly at different levels of intensity to normalize the impact of study we propose robustification level $\alpha$ which allows us to select the parameters of all perturbations such that, the performance drops to a specific level for all the perturbations i.e. $\frac{\#\text{of } \{f(x_n) \neq f(\zeta_n^t(x_n))\}}{\#\text{of } \{x_n\}} = \alpha$.

## 3.1 ROBUSTIFICATION.

We consider Cross entropy loss as the standard training loss $\mathcal{L}_s$ of a neural network with parameters $\theta$ trained on the training set $S = \{(x_n, y_n)|x_n \in X, y_n \in Y\}$.

**Adversarial Perturbations Training.** Next, considering the adversarial training method from Goodfellow et al. (2014) the network is trained with clean as well as with adversarial perturbed images. Hence, the total loss to optimize becomes $\mathcal{L}^\delta = \mathcal{L}_s + \mathcal{L}_r^\delta$. Adversarial loss is given as

$$\mathcal{L}_r^\delta = \min_\theta \frac{1}{|S|} \sum_{(\zeta_n^A(x_n), y_n) \in S} \mathcal{L}(f(\zeta_n^A(x_n)), y_n) \tag{3}$$

In spite of the fact that clean images are well represented in the training set this results in the drop of performance on clean images Zhang et al. (2019a). Additionally, these perturbations are different from the naturally occurring perturbations and are rarely found in practice. Therefore, we utilize a simple but effective technique to train the network on naturally perturbed images. We argue that natural perturbations enhance the class boundary more precisely. Basically, two clean images may differ by an elastic deformation or occlusions and training a network on them therefore help the classifier to learn better.

**Natural Perturbations Training.** In this section, we train our network with clean and naturally perturbed samples. We test these robustified networks on clean, adversarial perturbed and naturally perturbed samples. The total loss to optimize is given as $\mathcal{L}^\zeta = \mathcal{L}_s + \mathcal{L}_r^\zeta$. The loss for naturally perturbed images is

$$\mathcal{L}_r^\zeta = \min_\theta \frac{1}{|S|} \sum_{(\zeta_n^t(x_n), y_n) \in S} \mathcal{L}(f(\zeta_n^t(x_n)), y_n) \tag{4}$$

## 3.2 IMPLEMENTATION DETAILS

**Evaluation Metric.** In order to evaluate the performance of the classifiers we consider drop in the accuracy as an evaluation metric defined as

$$\Delta\mathcal{A} = \mathbb{1}(f(\zeta^t(x_n)) = y_n) - \mathbb{1}(f(x_n) = y_n) \tag{5}$$

where, $\mathbb{1}$ is the indicator function.

**Standard and Perturbed Image Classification.** We perform clean image classification using Resnet-152 network pre-trained on Imagenet and fine tuned on the respective dataset. We construct adversarial images using BIM method with number of steps $K = 10$ and $\epsilon$ values such that our desired drop in the accuracy is achieved as shown in Figure. 2 for each dataset. The similarity metric to determine similarity between clean and adversarial perturbed images is $l_\infty$ norm. We also construct naturally perturbed images using the method described in section 3. The parameters for natural perturbations are set in such a way that the same drop as adversarial is met for each dataset Figure.2 .

**Standardizing Comparison Among Perturbations.** We standardize the comparison among adversarial and natural perturbations by satisfying following two properties. 1) We select the parameters of the perturbations such that there is significant drop in the accuracy to study the robustness and images are still visually recognizable by humans. 2) for the fair comparison between adversarial and natural perturbations, the parameters of both types of perturbations are set such that the drop in the accuracy with natural perturbations is same as with adversarial perturbations for each dataset Figure. 2.

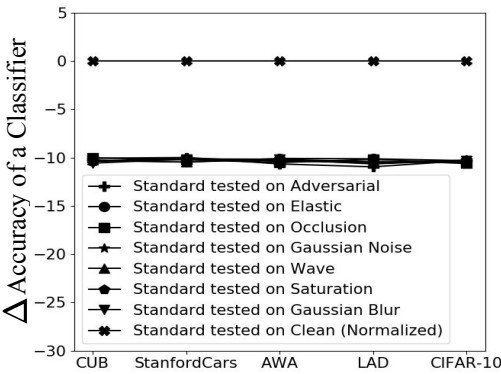

Figure 2: Calibrating the drop in the accuracy.

**Adversarial Perturbations Training.** We robustify the network using projected gradient descent method with the same $\epsilon$ values which lead to the drop in Figure. 2 for each dataset and number of steps $K$ taken as 10.

**Natural Perturbations Training.** We robustify the network with natural perturbations introduced in the images with the method explained in the section 3.1 and parameters which lead to drop in Figure.2.

## 4    EXPERIMENTS AND RESULTS

**Datasets.** The five datasets of varying size and granularity used in our experiments are Large Attribute Dataset (LAD) Zhao et al. (2018), Animals with Attributes (AwA) Xian et al. (2019), Stanford Cars dataset Krause et al. (2013), CUB-birds (CUB) Welinder et al. (2010) and CIFAR10 dataset Krizhevsky et al. (2009). LAD contains 78017 total number of images with 230 classes. We use (11702 train/ 9947 val / 9284 test) for our experiments. AWA contains 37322 images with 50 classes. We use (10450 train/ 7524 val / 9674 test). The CUB dataset consists of 11,788 images (5395 train / 599 val / 5794 test) belonging to 200 fine-grained categories of birds. Stanford Cars dataset contains (8144 train, 8041 test) images with 196 fine grained categories of cars. CIFAR10 dataset consists of 10 coarse grained categories with (50,000 train, 10,000 test) images.

### 4.1    STANDARD NETWORK CLASSIFICATION

**Normalizing Accuracy.** We start with evaluating the performance of a standard classifier on the clean images. A standard classifier shows the test accuracy of $81.20\%$ on CUB, $86.48\%$ on stanford Cars, $94.79\%$ on AwA, $83.75\%$ on LAD and $87.86\%$ on CIFAR10 dataset. For fair comparison among the results of different datasets we normalize the performance of the classifier on each dataset and show it with black line (cross symbol) on 0 in Figure. 2.

**Calibrating the Drop in the Accuracy.** Considering the accuracy of a standard classifier on the clean images as the baseline we drop the accuracy of the network with adversarial as well as natural perturbations to $10\%$. The maximum variation among drops with all the perturbations on one dataset is of standard deviation 0.26 which is negligible as compared to 10. The drop in the accuracy of the network with adversarial as well as natural perturbations is shown in Figure.2 (black lines at $-10$). Each black line shows the drop in the accuracy with a different type of perturbation and each point shows one experiment. We can observe from the plot that the drop for each dataset and each perturbation is achieved with minor variations among them. Hence, we achieve equal drop for both adversarial and natural perturbations to perform the fair comparison among them.

### 4.2    EVALUATING ROBUSTIFIED NETWORKS ON CLEAN IMAGES

We robustify the classifiers through adversarial training and natural perturbations training and contrast their performance on clean images. Figure.3a shows the performance for the clean images when the networks are robustified with adversarial training. The yellow line with the cross symbol in the

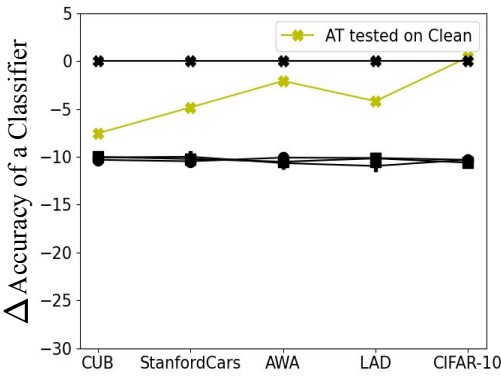 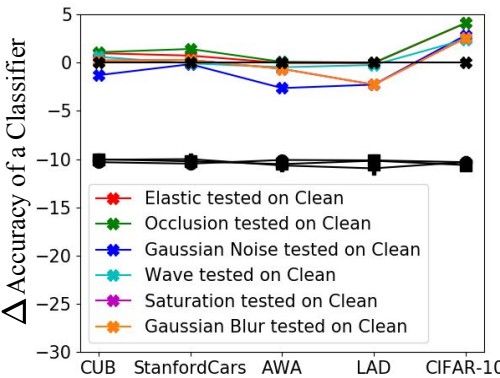

(a) **Evaluating adversarial training on clean images.**

(b) **Evaluating natural perturbations training on clean images.**

Figure 3: Comparing the performance of adversarial training with natural perturbations training on clean images.

plot shows the performance of the network robustified with adversarial training on clean images. While Figure.3b shows the performance when the network is robustified with natural perturbations and tested on clean images. Each line plot with a different color shows a network robustified on a different natural perturbation and tested on clean images (cross symbol). We observe that, adversarial training leads to drop in the accuracy on clean images for all the datasets except CIFAR10 while robustification with natural perturbations does not lead to the drop in the performance on the clean images. The network robustified with Gaussian perturbations for CUB, AWA and LAD dataset and the network robustified with Gaussian blur for LAD dataset does not completely recover the accuracy, however the drop is less as compared to robustification with adversarial training. For coarse grained CIFAR10 dataset robustification with all the natural perturbations even leads to improvement in the performance on clean images . Hence, this shows that robustification with natural transforms does not deteriorate the performance of network on clean images while robustification with adversarial perturbations leads to drop in the accuracy for clean images on four out of five datasets.

### 4.3 EVALUATING ROBUSTIFIED NETWORKS ON SEEN PERTURBATIONS

We compare the performance of adversarially trained networks with naturally robustified networks on seen perturbations. The results for the adversarially trained network tested on adversarial perturbations are presented in Figure.4a (yellow line with plus symbol). While, results for the naturally robustified networks tested on the same type of natural perturbations is shown in Figure.4b. Each line in the plot with a different color shows a network robustified on a different natural perturbation and tested on same kind of natural perturbation. By testing the performance of an adversarially trained network on adversarial images, and the naturally robustified networks on images containing same type of natural perturbations we observe that although adversarial training helps against adversarial perturbations however, the improvement in the performance with natural perturbations is higher. We further notice that for CIFAR10 dataset the drop in the performance due to natural perturbations is completely recovered. Hence, these results show that, robustification with natural perturbations outperform robustification with adversarial perturbations on the seen test set.

### 4.4 EVALUATING ROBUSTIFIED NETWORKS ON UNSEEN PERTURBATIONS

**Ineffectiveness of Adversarial Training Against Natural Perturbations.** The results for adversarially trained network tested on unseen natural perturbations are shown in Figure.5a, 5c, 5e. Each line in each subplot shows an adversarially trained network tested on a different natural perturbation with the symbol representing the type of the perturbation. We can clearly observe from the plots that adversarial training does not help against natural perturbations but it causes a further drop in the performance. This drop can even double e.g. against occlusion on the CUB, elastic and occlusion on AWA and Gaussian on LAD dataset. This is in contrast to the results presented in Laugros et al. (2019) where the authors showed that the performance of an adversarially trained network is

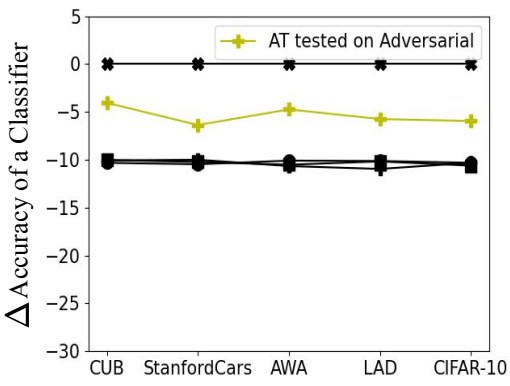 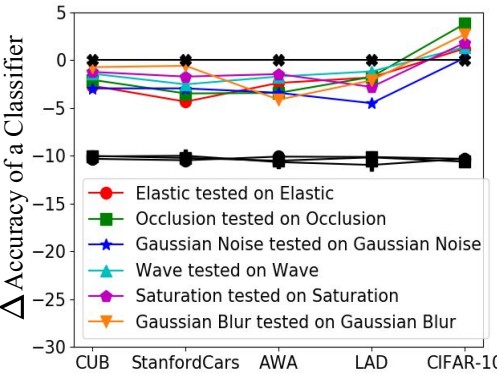

(a) **Evaluating adversarial training on the seen perturbations.**

(b) **Evaluating natural perturbations training on seen perturbations.**

Figure 4: Comparing the performance of adversarial training with natural perturbations training on seen perturbations.

the same as a standard network on natural perturbations. We argue that this difference is because we compare natural perturbations with adversarial after standardizing their effect. Thus, our results show that adversarial training does not generalize to natural perturbations but leads to further drop in the performance.

**Effectiveness of Natural Perturbations Training Against Adversarial and Unseen Natural Perturbations.** Each subplot in Figure.5b, 5d, 5f shows the results for a network trained on a different type of natural perturbation and tested on unseen perturbations. Within one subplot each line plot shows testing on a different perturbation with color representing the training perturbation and symbol representing the test perturbation. For example, the first subplot is trained on elastic perturbation (red color) and tested on occlusion , Gaussian noise, wave, saturation and Gaussian blur with symbols square, star, triangle up, pentagon and triangle down respectively.

In each subplot of Figure.5b, 5d, 5f line with "plus" symbol shows the performance for training with the natural perturbations and tested on adversarial perturbations. We can clearly observe that robustification with natural perturbations generalizes to adversarial perturbations. Furthermore, this shows a similar pattern among all the six natural perturbations training. On CUB and StanfordCars dataset it recovers all the drop, on AWA and LAD datasets it reduces the drop from $10\%$ to around $2\% \sim 3\%$, and on CIFAR10 dataset it even helps to improve the accuracy on adversarial images. Contrary to this in Laugros et al. (2019) the authors depicted that training with natural does not generalizes to adversarial. Our results show that, robustification with natural perturbations training transfers to adversarial perturbations.

Augmentation with elastic perturbations leads to improvement in the performance against all the unseen natural as well as adversarial perturbations except Gaussian perturbation. For CIFAR10 it also leads to drop on wave perturbation besides Gaussian. Elastic and wave which look similar however, they do not perform well on each other on CIFAR10 which shows that they are not correlated on CIFAR10. Training with occlusion perturbations shows a similar behavior as elastic it also enhances the accuracy on all the unseen perturbations except Gaussian and wave perturbations on the CIFAR10 dataset. However, it significantly helps to improve the performance against elastic and saturation perturbations on the CIFAR10 dataset.

Training with Gaussian perturbations provides minimum defense against unseen natural perturbations and shows worst performance on AWA dataset. On AWA dataset it leads to a further drop in the performance from $10\%$. Augmentation with saturation does not lead to much drop on unseen test set however it does not improve much either. Robustification with the wave perturbations helps against all of the unseen perturbations except a little drop on Gaussian perturbation for CIFAR10 dataset. Finally, training with the Gaussian blur shows an average behavior, for some perturbations like elastic on CIFAR10 it leads to the complete recovery in the drop whereas for some of them like Gaussian noise it leads to the further drop in the performance and for rest of the unseen perturbations it did not help. We also observe that, robustification with any of the natural perturbations transfers to adversarial however most of them fail to perform well on the Gaussian noise this implies that

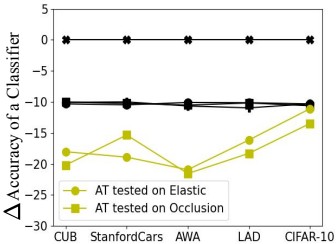

(a) **Evaluating adversarial training on the elastic and occlusion perturbations.**

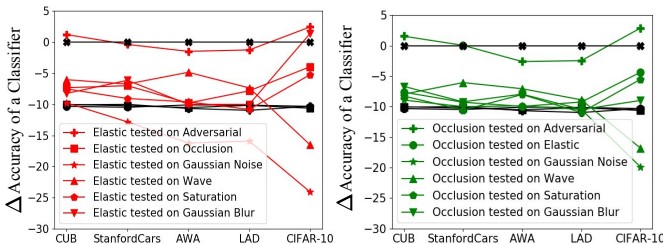

(b) **Evaluating natural perturbations training on unseen perturbations.**

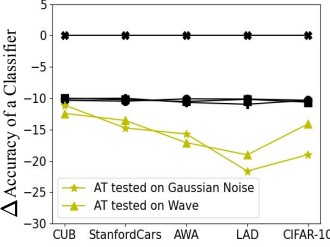

(c) **Evaluating adversarial training on the Gaussian noise and wave perturbations.**

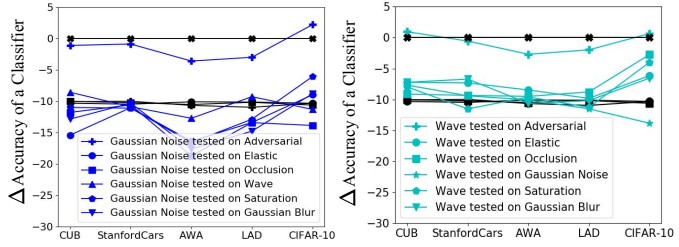

(d) **Evaluating natural perturbations training on unseen perturbations.**

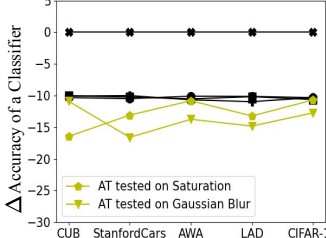

(e) **Evaluating adversarial training on the saturation and Gaussian blur perturbations.**

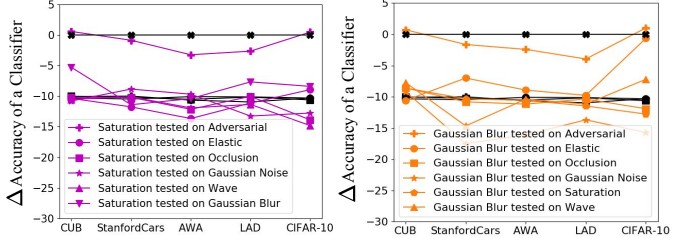

(f) **Evaluating natural perturbations training on unseen perturbations.**

Figure 5: Comparing the performance of adversarial training on unseen perturbations with natural perturbations training on unseen perturbations.

adversarial and Gaussian are not correlated. Thus, evaluation on unseen perturbations depict that augmentation with all of the six natural perturbations under consideration robustify the networks against adversarial perturbations. By comparing the subplots with each other we learn that "elastic", "occlusion" and the "wave" are the best performing ones.

## 5 CONCLUSION

In this work, we focus on general robustness and robustify networks with natural as well as adversarial perturbations while standardizing comparisons among them. We demonstrate that adversarial training leads to the drop in the accuracy on clean images while robustification with natural perturbations does not degrade the performance on the clean images, even for CIFAR10 it leads to the improvement in the performance. We also showed that classifiers trained with natural perturbations show better improvement in the performance on seen perturbations than adversarial training on adversarial images. Finally, we contrasted the results of adversarially trained networks on unseen perturbations, with natural perturbations trained networks on unseen perturbations. We observed that all the natural perturbations being considered improved the accuracy on adversarial examples. "Elastic", "occlusion" and "wave" showed the best performance on unseen perturbations. In contrast, adversarial training lead to a further drop in the accuracy on unseen perturbations. Thus, although general robustness against any arbitrary perturbation is hard to prove, we conclude that natural perturbations added to the training scheme provide always better robustness than adversarial training does to (almost) any of the unseen perturbations we have provided.

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

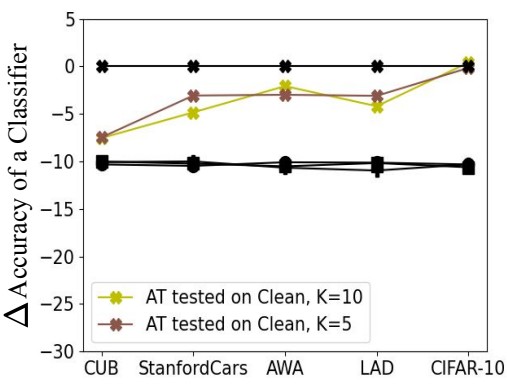 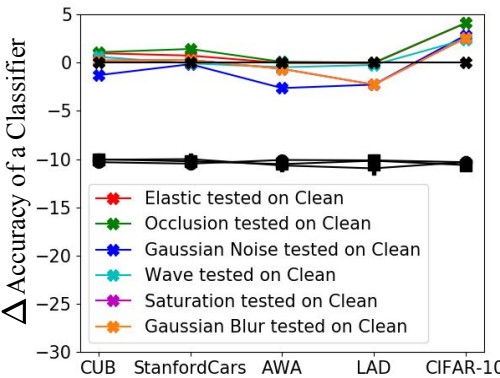

(a) **Evaluating adversarial training on clean images.**

(b) **Evaluating natural perturbations training on clean images.**

Figure 6: Comparing the performance of adversarial training with natural perturbations training on clean images.

# A    APPENDIX

## A.1    NATURAL PERTURBATIONS TRAINING ALGORITHM

---
**Algorithm 1** Natural Perturbations-based Training for Robustification.

---
1: Given $S = \{(x_n, y_n) | x_n \in \mathbb{R}^2, y_n \in \mathbb{N}\}$. Learning rate $\eta$. A set of natural perturbations $\zeta^t$.
2: Initialize $\theta$ randomly
3: **for** $epoch = 1$ to $N_{ep}$ **do**
4:     **for** $minibatch\ B \subset |S|$ **do**
5:         $\mathcal{L}_s = \mathcal{L}(f(x_n), y_n, \theta)$
6:         **if** $epoch > delay$ **then**
7:             $\mathcal{L}_r^\zeta = \mathcal{L}(f(\zeta_n^t(x_n)), y_n, \theta)$
8:             $\mathcal{L}^\zeta = \frac{\mathcal{L}_s + \mathcal{L}_r^\zeta}{2}$
9:         **end if**
10:         Update $\theta$ with SGD.
11:         $\theta = \theta - \eta \bigtriangledown_\theta \mathcal{L}^\zeta$
12:     **end for**
13: **end for**

---

## A.2    EVALUATION FOR VARYING PARAMETERS OF ADVERSARIAL PERTURBATIONS.

In this section we present the results for adversarial perturbations generated with $K = 5$ and $\epsilon$ adjusted such that the same drop of $10\%$ is retained for the fair comparison among perturbations.

**Evaluating Robustified Networks on Clean Images.** In Figure.6a we observe that by varying the number of steps required to generate adversarial examples from $K = 10$ to $K = 5$ and the perturbation size while keeping the drop same, the performance of the adversarially trained network does not vary significantly on clean images. By comparing the plot for $K = 5$ in the Figure.6a with the plots in the Figure.6b we learn that, training with natural perturbations provides better recovery in the drop of performance on the clean images better than the network adversarially trained with $K = 5$. Hence, our results show that, by varying the number of steps and perturbation level for generating adversarial examples while maintaining the drop, the behavior of an adversarially trained network on the clean images does not change significantly.

**Evaluating Robustified Networks on Seen Perturbations.** Figure.7a shows the results for two adversarially trained networks on adversarial examples with different parameter configurations while keeping the drop to $10\%$. We observe that with the change in the parameters of adversarial training

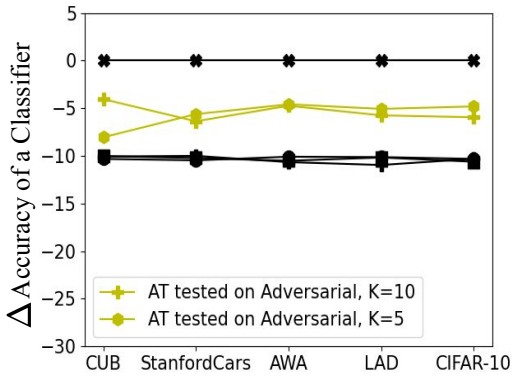 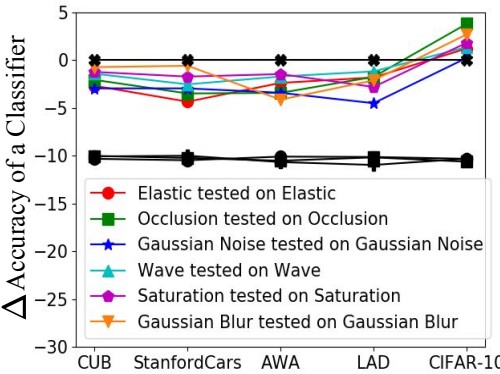

(a) **Evaluating adversarial training on the seen perturbations.**

(b) **Evaluating natural perturbations training on seen perturbations.**

Figure 7: Comparing the performance of adversarial training with natural perturbations training on seen perturbations.

the performance on the adversarial examples does not vary significantly. Only for CUB dataset the network with number of steps $K = 10$ performed better than $K = 5$. The contrast between the plots in Figure.7a and Figure.7b depicts that training with natural perturbations transfers to natural better than adversarial training on adversarial perturbations. Therefore, we learn that by varying the number of steps and perturbation level for generating adversarial examples while maintaining the drop, the behavior of an adversarially trained network on the adversarial images does not change significantly.

**Evaluating Robustified Networks on Unseen Perturbations.** Figure.8 (left) shows the results for two adversarially trained networks with different parameter configurations on unseen perturbations. Figure.8 (right) shows the plots for networks trained on natural perturbations and tested on unseen perturbations. Each subplot on the Figure.8 (right) shows a network trained on a different type of natural perturbation and tested on the unseen perturbations.

By contrasting yellow line plots (for $K = 10$) with brown line plots (for $K = 5$) in Figures.8a, 8c, 8e we observe that the performance of adversarially trained networks with $K = 5$ and $K = 10$ on unseen natural perturbations does not vary significantly. We notice the difference in performance only on CUB dataset among two networks. An adversarially trained network with $K = 5$ for CUB dataset shows better recovery against elastic perturbations than $K = 10$ network. However, it shows worst performance against Gaussian noise, wave and saturation on the CUB dataset.

The line plots in each subplot in Figures. 8b, 8d, 8f with symbols "plus" and "hexagon" show the performance of naturally trained networks on adversarial perturbations with $K = 10$ and $K = 5$ respectively. By contrasting their performances in each subplot in Figure.8 (right) we observe that the recovery with the natural perturbations is almost the same except some differences on the CUB dataset. We observe that for the CUB dataset all the natural perturbations trained networks recovered the drop against adversarial examples with $K = 10$ steps better than $K = 5$. Thus, our analysis on unseen perturbations show that with the change in the parameters of adversarial perturbations while keeping the drop same overall there is no significant change in the performance of networks both adversarially trained as well as natural perturbations trained. However, for fine grained CUB dataset we observed that an adversarially trained network with $K = 10$ is better at unseen robustification than $K = 5$. On the other hand, natural perturbations trained networks are better at recovery against $K = 10$ adversarial perturbations on the CUB dataset.

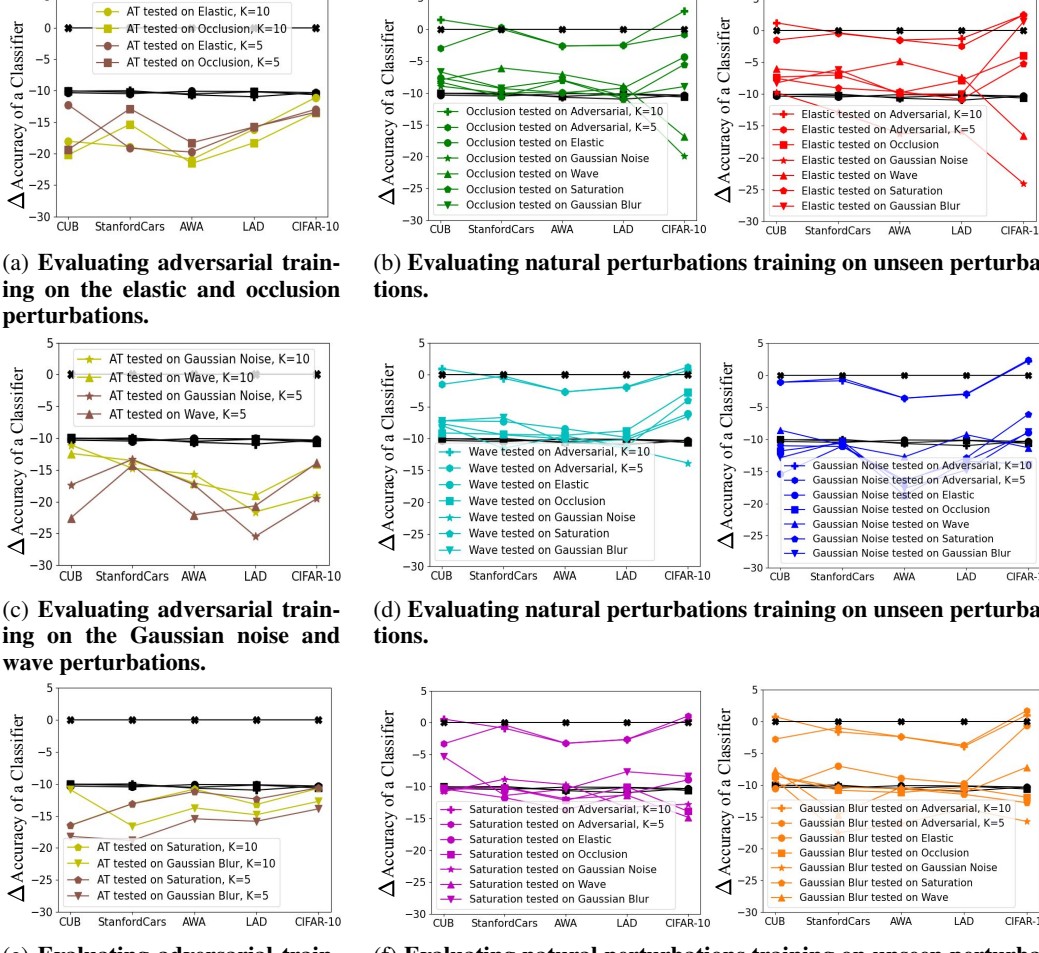

(a) **Evaluating adversarial training on the elastic and occlusion perturbations.**

(b) **Evaluating natural perturbations training on unseen perturbations.**

(c) **Evaluating adversarial training on the Gaussian noise and wave perturbations.**

(d) **Evaluating natural perturbations training on unseen perturbations.**

(e) **Evaluating adversarial training on the saturation and Gaussian blur perturbations.**

(f) **Evaluating natural perturbations training on unseen perturbations.**

Figure 8: Comparing the performance of adversarial training on unseen perturbations with natural perturbations training on unseen perturbations.

