# OpenReview forum: "Adversarial and Natural Perturbations for General Robustness"
_ICLR.cc/2021/Conference — Reject_

### Official Review · AnonReviewer4 · 2020-10-28
**An important problem, but the results are hard to interpret**

**Rating:** 4
**Confidence:** 5

**Review:**

#### Summary

In this paper, the authors evaluate the performance of classifiers trained and then later tested on both adversarially generated perturbations as well as more natural perturbations.  By considering six different natural perturbations, they show empirically that natural perturbations can improve performance against clean and adversarially-perturbed images.  They also show that adversarial training does not improve performance on unseen natural transformations.

#### Strengths

- This paper studies an interesting problem.  Natural perturbations have gained increasing interest from the robustness community recently.  While there are many works that study the trade-offs (e.g. in the test accuracy of clean and adversarialy perturbed data) with respect to adversarial perturbations, these trade-offs are much less well understood for more "natural" notions of robustness.

- The performance normalization step here is interesting, and in my opinion is an useful way to compare notions of robustness across domains.  As this problem receives more attention, I believe that this will be a useful metric for comparing results for the community.

- The result that (in some cases) adversarial training hurts natural robustness and natural robustness helps adversarial robustness is interesting and noteworthy.  A more in-depth analysis of these phenomena and whether they hold for broader classes of natural transformations would be quite interesting.

#### Weaknesses

- Although the title advertises that this paper considers "natural" perturbations of data, it is arguable as to whether these transformations are representative of transformations that are more likely to be "found in the real world" (quoted from the Introduction).  Although it seems true that adversarial perturbations may not occur naturally, I'm not entirely convinced that these transformations are any more likely to be encountered.  Moreover, based on Figure 1, the elastic, wave, gaussian noise, and gaussian blur transforms all look quite similar to one another.  A more convincing argument could be made here if a more diverse set of realistic perturbations was considered  (e.g. changing the weather conditions in iamges).

- The notation throughout this paper is at times confusing.  I bullet the instances where I felt the notation was unclear:

   + it seems unintuitive as to why $\zeta_n^A$ and $\zeta_n^t$ need to depend on $n$.
   + The authors write that each image $x_n$ is an element of $\mathbb{R}^2$ -- I assume this is a typo.
   + The symbol $\circledast$ is undefined -- is this a pointwise convolution?
   + The constant $\alpha$ is not defined in the definition of $\zeta^O$, and the notation $b^{x_c, t,r}$ is confusing -- I'm still unsure of how $\zeta^O = \min(x_n, b^{x_c,t,r})$ would be applied in practice.  What is the size of $b$?
   + The notation $x_n^{N(\mu,\sigma^2)}$ is confusing -- I assume this means that $x_n \sim N(\mu,\sigma^2)$?
   + The "shift operator" in the definition of the wave transformation is also undefined.
   + In the definition of the accuracy drop, the symbol $\alpha$ is reused; previously, it was also used to denote parts of $\zeta_S$ and $\zeta^E$.  To this end, the definition of $\alpha$ here is confusing -- shouldn't it be defined in terms of the cardinality of these sets?
   + Why is the cross entropy loss indexed by lower-case $s$, which seems to be undefined?

- The figures are quite hard to parse.  In particular, although the message of Fig. 2 is clear from context, the symbols all overlap, which when we look at later figures makes hard to interpret.  Also, it's unclear why all the lines are connected, as each data point on the line simply corresponds to a different dataset.

- In some sense, the fact that natural transformations improve clean accuracy is already known.  The training scheme that is introduced for training using natural perturbations is simply data-augmentation.  And many works have previously shown that data-augmentation (e.g. [1], [2], [3]) improves clean accuracy.

- I'm a bit surprised that adversarial training does not decrease the performance on clean images (e.g. Fig 3a) for CIFAR-10.  Unless I have misunderstood, this is at odds with other works (e.g. [4]) that claim that robustness is at odds with accuracy.

- I'm not sure how to interpret Fig 5.  In particular, it seems that in some cases, training with data augmentation improves performance and in others, performance suffers when evaluating on unseen transformations.  This makes the claim of the paper feel somewhat weaker, given that the message emphasized in the intro only holds for a subset of the combinations of training/testing transformations.

#### Final thoughts

While I think that this paper studies an interesting problem, and has some interesting ideas such as normalizing the accuracy drop to create a fairer comparison, I feel that the experiments don't fully support the claimed conclusion.  Specifically, the phenomena of interest, such as that data augmentation improves performance on unseen natural transformations and on adversarially perturbed inputs, seems to not hold in general; on the contrary, it appears that a non-negligible amount of the time, it also hurts performance.  Further, the presentation is hard to follow at times due to grammatical mistakes and undefined notation.  Also, the natural transformations under consideration are all quite similar -- it would be very interesting to see if these kinds of results would hold with respect to much more severe changes in images, such as adding snow or changing the background colors.  For these reasons, I am leaning toward suggesting rejection for this paper.

#### References

[1] https://link.springer.com/article/10.1186/s40537-019-0197-0
[2] https://ieeexplore.ieee.org/document/8388338
[3] https://arxiv.org/pdf/1708.04896.pdf
[4] https://arxiv.org/pdf/1805.12152.pdf

---

### Official Review · AnonReviewer3 · 2020-10-28
**Anonymous Review**

**Rating:** 4
**Confidence:** 4

**Review:**

Paper summary
This paper studies the degradation in performance of deep learning models under both adversarial as well as natural perturbations or contamination. It proposes to study and compare their effects, as well as their potential use for defenses across different scenarios.

Review Summary
To this reviewer, the questions that these papers asks are somewhat interesting as the understanding of the robustness of machine learning models in general is an important questions. However, I believe the reported results are naturally expected and the paper is hard to read.

Detail Review

Strengths:
- This paper studies a relevant problem
- There is a large empirical study of different training and testing settings.

Weaknesses
- The driving question of this paper seems to be whether natural perturbations hurt generalization as much as adversarial examples. The authors answer this question in the negative, but this is naturally expected: the perturbations that the authors study yield images that are likely in the distribution of natural images. As a result, rather than perturbations, they could well be considered as data augmentation.
- On section 3 (introducing the updates in Eq (1&2)) the authors write "the equation to be optimized is given as". However, what follows is an equation, and not any optimization problem to be optimized.
- The authors stress that natural perturbations are designed with care, so that they induce a control loss in accuracy while being similar enough to the original samples so that they are recognizable by humans. How is this verified? The occlusions are drawn at random, so could it not be the case that the main object in the image is occluded?
- The expression for the perturbed samples with Gaussian noise makes no sense to me. I believe this should read $\zeta(x_n) = x_n + v$, $v\sim N(\mu,\sigma^2)$.
- It is not always clear if the reported results are over training or testing samples. For example, at the end of Sec 4.3, the authors comment "robustification with natural perturbations outperform robustification with adversarial perturbation on the seen test set". Some other times, the authors refer to "the unseen test set". What does it mean that the test set is 'seen' or 'unseen'?
- To make comparisons fair, the authors calibrate the energy of all perturbations. However, the definition of this quantity $\alpha$ simply measures whether there exist a perturbation on $f(x_n)$ or not, regardless of whether $f(x_n)$ was a correct prediction. In other words, are the authors assuming a perfect model?
- The plots are a bit hard to interpret, as one needs to distinguish between overlapping symbols of the same color (see e.g. Fig 5.b).
- In defining the evaluation metric, $\Delta \mathcal A$, the authors must have meant the average of the reported quantity over the samples.
- There authors refer to Carlini & Wagner (2017) as proposing adversarial training, in the beginning of section 2. I believe this is incorrect, and that they must have meant to cite Goodfellow et al. (2014) or Madry et al (2017)
- The paper is hard to read and English usage should be improved. I read the paper a few times, and I'm hoping I understood the authors claims and results correctly (which I believe I did).

---

### Official Review · AnonReviewer1 · 2020-10-29
**Potentially interesting ideas but unclear contributions**

**Rating:** 4
**Confidence:** 4

**Review:**

This paper studies the effect of “robustification” (i.e., adversarial training or data augmentation) of models on the accuracy to seen and unseen perturbations. The authors propose a technique to “standardize” the robustification process across different perturbations. They evaluate their approach on several datasets, highlighting how standardization yields different insights compared to prior work.

Pros
- Standardization is an interesting idea

Cons
- Unclear contributions, and generally poor clarity and presentation
- No justification of standardization technique
- Experiments do not study dependence on standardization level

It is unclear exactly what contributions this paper is making. First, as the authors acknowledge, there have been several recent works studying robustness to natural vs. adversarial perturbations. For completeness, one the authors miss is the following:

Taori et al., Measuring Robustness to Natural Distribution Shifts in Image Classification. On arXiv.

In more detail, the authors study “robustification” of the DNNs  on adversarial vs. natural examples. For natural examples, this is really just standard data augmentation, whereas for adversarial examples it is adversarial training.

It appears that the main difference compared to prior work, Laugros et al 2019, is that the authors “standardize” the robustification by capping the permitted decrease in robustness. This is an interesting concept, though the authors need to do a significantly better job explaining and justifying it. For instance, the authors claim that their results differ from Laugros et al 2019 due to standardization, but it is not clear why standardization would cause these differences.

More generally, it is not really justified why standardization should be important. Intuitively, it makes sense to me that different kinds of perturbations may not be directly comparable due to different levels of shifts. However, there is no reason why standardization is the right way to normalize them. For instance, an alternative would be to tune the level of the shift based on the drop in accuracy of the *unrobustified* network. Why should the authors’ standardization technique better?

Equally importantly, the authors do not study the choice of different standardization levels on their results. Given the apparent importance of standardization to the authors’ contributions, they should justify that their results are not dependent on specific hyperparrameter choices (or at least study how the hyperparameter of standardization level affects their results).

As written, it is not clear what meaningful insights the authors are studying. For example, they find that “robustification with natural perturbations outperform robustification with adversarial perturbations on the seen test set”, but this fact is unsurprising given that natural perturbations are (presumably) naturally present in the training data to some degree (though, depending on the transformation, the term “natural” is debatable).

Finally, the figures are not well explained very well, making it even harder to understand exactly what the authors are trying to convey. For instance, the authors should define give precise definitions of “\Delta Accuracy of a Classifier”, and preferably give a reminder in the captions. In addition, not all the curves in the figures are labeled. Also, it is unclear why the authors are using a line plot, which is confusing).

In summary, I think this paper has some interesting ideas, but it is far from ready for publication and the author need to spend significant effort clarifying their contributions and improving the clarify of their paper.

---

### Decision · Program_Chairs · 2021-01-07
**Final Decision**

**Decision:**

Reject

**Comment:**

The reviewers indicated a number of concerns (which I agree with) which have not been addressed by the authors as they have not provided any response.  Indeed, the paper would be significantly improved once these issues are addressed.